

# Modular inverse visual cryptography for balancing security, quality, and efficiency in image transmission

Selva Mary[1], John Blesswin[1], Suresh Sankaranarayanan[2] and Abdul Rahaman Wahab Sait[3]

[1] Directorate of Learning and Development, SRM Institute of Science and Technology, Kattankulathur, India
[2] Computer Science, King Faisal University, Al Hofuf, Saudi Arabia
[3] Documents and Archives, King Faisal University, Al Hofuf, Saudi Arabia

## ABSTRACT

In this article, a new Inverse Module Visual Cryptography (IMVC) system is proposed to transmit a small computed tomography (CT) scan image over the IoT network securely and efficiently. In particular, Internet of Things (IoT) devices are important for the healthcare industry as they transmit sensitive imaging data; thus, confidentiality and high image fidelity are critical. The IMVC framework stands as a blend between visual cryptography and the very efficient and light-weighted modular arithmetic, suitable for mostly computational-constrained IoT networks. In this way, CT scan images are divided into pixel-aligned shares, inserted into cover images, and securely sent to destinations of the receiver. Expected outcome shows that the IMVC reconstructs equally a complete and lossless image, which displays equal quality parameters such as an infinite peak signal to noise ratio (PSNR) and equal mean square error (MSE) of zero, signifying the reconstructed images were accurate. Furthermore, expected high immunity to brute-force, statistical, and collusion attacks makes the methodology more reliable for improving the security of the IoT-based medical application. As conceived, performance tests are expected to report an efficient computational building and time required to encrypt and decrypt the framework, which will amount to an average of 1.79 and 0.45 s, respectively, implying real-time suitability of the framework in clinical settings. In this work, the proposed IMVC allows for the development of a feasible and effective solution for secure transfer of important image data in an IoT healthcare context, while addressing the requirements of security, image quality, as well as efficient transmission.

# INTRODUCTION

In today's connected world the delivery of imagery across the Internet of Things (IoT) networks has become significant across numerous fields such as health, social media, business, and military. Due to the unprecedented IoT proliferation that has created an enabling environment for a continuous flow of data between connected gadgets, there is a considerable emphasis on ensuring the secure transfer of image data, in particular their confidentiality and protected integrity. IoT networks convey images that quite often

Corresponding authors
Selva Mary, selvam1@srmist.edu.in
Suresh Sankaranarayanan, ssuresh@kfu.edu.sa

contain important data, and hacking can cause huge losses, invasion of privacy or even compromise national security. In the healthcare industry, consumer IoT is used to capture, and transfer medical images including X-rays, magnetic resonance imaging (MRI), and computed tomography (CT) scans which are highly sensitive information about the patient. The privacy of these images while in transit should also be highly guarded so that unauthorised persons do not gain access to the patients' data (*Naor & Shamir, 1994*; *Ateniese et al., 1996*). These six concerns that were presented above are characteristic of each specific IoT application of the image communication. For instance, the position of images in social media applications aide in ensuring user privacy and checking on identity theft (*Aswad, Salman & Mostafa, 2021*). Security of the images in military IoT application is vital in that it helps protect sensitive information that is strategic to any nation thereby serving the national interest (*Sambas et al., 2020*). In this healthcare domain, medical images are fundamental in diagnostic imaging and patient management, and flawed images can cause diagnostic mistakes and treatment errors and compromise the confidentiality of the patient (*Rong et al., 2024*). Therefore, as reflected in works (*Akanksha, Garg & Shivani, 2023*), protection of these images in IoT environments is not only a concern of privacy but also the quality and accuracy that is necessary for medical evaluation. Classic cryptogram approaches, mainly initially created for textual data security, come across great difficulties if applied to image in IoT systems. In contrast with text, images are characterized by the presence of a sheer amount of data which has built-in redundancy and, hence, is not so easy to encrypt (*Çiftci & Sümer, 2022*; *Sambas et al., 2022*). Conventional techniques of cryptography applied to images, cause such problems as pixel enlargement which leads the encrypted image's size to be larger than the original image which results in inefficient storage and transmission—a drawback prohibited in bandwidth-restricted IoT networks (*Sankaranarayanan et al., 2024*). In addition, traditional encryption methods may cause a significant loss in picture quality, which is highly undesirable in IoT health care applications using pictures where picture clarity and sharpness often dictate the detection of diseases (*Liu et al., 2024*; *Pan et al., 2021*). Visual cryptography (VC), developed by *Naor & Shamir (1994)*, offers a new solution exclusively for images (*Zhang et al., 2023*). VC enables a secret image $X_S$ to be decomposed into several shares $T_i$, such that none of the shares can disclose any information about the others. Complete image can be reconstructed only when sufficient number of shares are combined and as such, VC is an effective tool for secure image transmission. However, to reconstruct the image, the shares are superimposed by using the human visual system (*Rani, Sharma & Mishra, 2022*; *Salim, Abboud & Yildirim, 2022*). Although expansion can be very effective, this expansion can lead to lower resolution and quality of the reconstructed image (*Naor & Shamir, 1994*). To address the drawbacks of binary VC, grayscale VC was proposed, which permitted each pixel to contain multiple intensities rather than just two; thus, they retain much more information from the input image than binary VC and are more appropriate for use in high definition systems, such as IoT-based diagnosis in healthcare facilities. Therefore, there are still some obstacles to overcome them like pixel expansion and quality of the reconstructed image (*Sherine et al., 2022*; *Wang et al., 2020*; *Yang et al., 2025*; *Sambas et al., 2024a*, *2024b*). Color VC schemes have to meet the following criteria: The image

reconstruction achieved by the method must be as good as possible, whereas the computational complexity has to be considered due to the IoT applications (*Wang et al., 2023*).

However, some problems are still on the way of implementation of VC: A big drawback of the classical approach to the implementation of VC techniques is the loss of image quality during the reconstruction. Due to the binary form of the early VC schemes, the amount of information loss is high, especially for applications that require high accuracy, such as medical imaging (*Karunya & Mary, 2025*). In general grayscale and color VC, the problem of preserving the identity of the reconstructed image remains a major issue (*Yang & Laih, 2000*). Many of the conventional approaches have the problem of pixel expansion in which the size of the encrypted shares is greater than the image itself. This not only incurs storage overheads but also makes the passage of share information over the network even more challenging especially in environments that are constrained by bandwidth (*Rajpurkar et al., 2018*). The processes of generating shares and reconstruction of the original image may require considerable number of computational steps where large color images are involved. This makes it difficult to use VC in real-time application or systems that are restricted in terms of resources (*Çiftci & Sümer, 2022*). Different approaches have been suggested to address these challenges; however, the improvement of security, image quality and reducing the amount of computations simultaneously is still a main concern (*Mary, Blesswin & Kumar, 2022*).

A fundamental operation in the number theory is the inverse modulo arithmetic (IMA), which is used in this research. A number $b$ is called the modular inverse of a number $a$ under a modulus $m$ if $a \times b \equiv 1 \ (mod \ m)$. In public key cryptography systems, the security of the decryption operation can rely on the computational difficulty of computing modular inverses without the private key (*Liu et al., 2024*; *Long et al., 2023*; *Blesswin, Mary & Kumar, 2021*), which property is crucial. IMA is about image security in the IoT networks, by maintaining images integrity and confidentiality without pixel expansion. Then, the IMA applies encryption and decryption inside the fixed modular range to reduce the risk of overflow or expansion, so that the encrypted image has the same size as the original (https://nihcc.app.box.com/v/ChestXray-NIHCC/folder/36938765345). Nevertheless, application of IMA in image encryption for IoT scenarios faces problems, in particular preserving image quality and decreasing computational overhead, especially in consideration of the aforementioned limited availability of computing resources for IoT devices.

The source of the motivation for the IMVC approach stems from the inefficiencies evident in normal VC techniques when implemented in the restricted environment of the IoT networks. But, traditional VC schemes, suffer from certain significant problems such as pixel expansion, reduction of image quality, and computational complexity. In any application of CT scan images in healthcare, it is critical to maintain high image quality because any distortion is likely to have a straight influence on the medical diagnosis of patients' ailments. Furthermore, the usual pixel expand, which is characteristic of classical VC, not only raises the transmission burden but also limits the applicability of these methods in real-time transmission of medical data. These limitations suggest a new

strategy that can cope with these difficulties, but at the same time, is feasible for the devices with the limited computational power.

The IMVC methodology adds an adaptive modular arithmetic that prevents pixel expansion during both encryption and decryption phases. Through applying the modular inverse operations, IMVC retains the dimensions of the encrypted shares in a way that makes this approach useful for IoT systems where bandwidth is limited and where pixel expansion would be expensive. The application of modular inverse also ensures that shattered image quality is restored fully to that of the original image quality hence satisfying the requirements of medical imaging where the accuracy of diagnoses highly relies on image sharpness. The structure of this article is as follows: In 'Materials and Methods', the proposed IMVC methodology along with the encryption and decryption procedures are illustrated. 'Results' focuses on the system and its performance as well as the results from the implementation of the system. At the end of each section, 'Conclusions' presents a conclusion of the study along with the recommendation and possible future research.

## MATERIALS AND METHODS

In the situations when secure communication uses open and public networks, the confidentiality and integrity of the transferred images, including the medical ones, should be ensured. While applying conventional cryptographic methods, pixel expansion and increased computational load may be encountered as main problems, thus degrading the quality of the images and the security level of the system as well. To meet these challenges, the proposed IMVC methodology combines VC and modular arithmetic, incorporating a new stable approach that improves security, yet reduces computation time, with no significant deterioration of image quality. The usefulness of the proposed IMVC approach in image security is rooted in its two-folded approach of improving image security, on the one hand, and improving efficiency of the encryption and decryption process, on the other hand, in a way that would be integrated with inverse modular arithmetic and VC. Another important improvement is that of the combination of least significant digit (LSD) embedding with the inverse of the modular arithmetic operation. This procedure hides message pixel values in visually sensible cover images so that it can hardly be detected while transmitting and offers double-layer security. While the shares generated by the IMVC process are random data points, these are generally contained within contextually meaningful images, which makes it less conspicuous to unauthorized viewers and improves steganographic security. Moreover, the proposed IMVC approach adds value to the existing methods by enhancing the computational load. The encryption process involves basic mathematical calculations, and the encryption plan does not include computationally intensive operations, and therefore it can be implemented in real-time conditions on healthcare IoT devices. In summary, the IMVC approach advances the state-of-the-art in VC by:

1. Pixel expanding is removed, bandwidth usage is conserved, and the shapes are preserved.

2. Molesting the reconstructed image in order to maintain high fidelity.

3. Incorporation of shares within cover images of correlated meaning for security.

4. Reducing the computational complexity required for implementation in real-time applications.

The proposed IMVC is structured into two critical phases, namely the share generation phase and the reconstruction phase. Using a combination of keys, the proposed IMVC performs the encryption and decryption process in reverse modulo. It is one of generating multiple share images ($T_i$) from a secret image ($X_S$) that prevent the data from being recovered by utilizing cover images ($C_i$) to hide the data and ensure that ($X'_S$) is reconstructed only by authorized entities with the right keys. It is efficient, secure, with minimal pixel expansion and the original image quality is not well impacted, suitable for applications that are both security and image fidelity critical.

The proposed IMVC utilizes three security keys: the encryption $aK, mK$ and key $eK$. The security of the is dependent on these keys. The encryption and decryption processes are done by $eK$. Later, for initial encryption of the $XS$, a Key $aK$ is used to further complicate the encrypted data, making it more $mK$ in conjunction with is resistant to unauthorized access. This assurance is provided by modular arithmetic, which keeps all operations within a fixed range, thereby avoiding overflow. As a result, the encrypted and decrypted data remain consistent (*Putranto et al., 2023*). The trusted third-party (TTP) storage stores keys system. A storage service or entity is referred to as TTP storage system. trusted by the parties who participate in a communication (*Liu et al., 2024*).

## Share generation phase

The share generation phase of the proposed is displayed in Fig. 1. In the share generation phase, allows the $XS$ to Keys $eK, aK,$ and $mK$ for the encryption processes. This The process not only secures the image, but also breaks it up into each share image $T_i$, $\forall\ i\ \in \{1, 2, 3\}$ contains a $i$ and use a portion of the encrypted data. The LSD algorithm covers these shares with $C_i, \forall\ i\ \in \{1, 2, 3\}$ authorized users are ensured that the during transmission (*Sankaranarayanan et al., 2024*), the secret image is secure.

The step by step process is shown as follows:

**Step 1: Encryption of secret image:**

The secret image $X_S$ is represented as a matrix with $M$ rows and $N$ number of columns. $X_S$ consists of pixel value in each pixel which is in a particular range commonly it varies 0 to 255.

The encryption of the secret image $X_S$ is performed using Eq. (1)

$$Z_E(x, y) = mod((eK \times X_S(x, y) + aK), mK) \tag{1}$$

where $Z_E(x, y)$ is the pixel value of the encrypted image at position $(x, y)$ and $0 \leq Z_E(x, y) \leq 255$. $X_S(x, y)$ is the pixel value of the secret image. Encryption key $eK$ is a scalar that scales the pixel values, additive key is a scalar $aK$ that shifts the pixel values, modulus key is $mK$ that defines the allowable range of the resultant pixel values and $mod(a, b)$ denotes the modulo operation. To properly encrypt and so as to prevent value overlap, the modulus key $mK$ must be chosen so that $mK > max(X_S)$. The selection of pixel

Share Generation Phase

**Figure 1 Share generation phase of the proposed IMVC.**

values within the acceptable range $[0, mK - 1]$ (*Sherine et al., 2022*) require careful selection of the encryption and additive keys $eK$ and $aK$.

**Step 2: Generation of shares:**

This encrypted image $Z_E$ of size $M \times N$ is partitioned into $n$ shares $(T_i)$ using a VC process. For each pixel $Z_E(x, y)$ in the encrypted image, the individual digit components $d_k$ are extracted according to their place values. These digits are then used to replace the LSDs in the corresponding pixels of the cover images $C_i(x, y)$ as follows.

*Digit extraction:* Each pixel $Z_E(x, y)$ is decomposed into constituent digits using Eqs. (2), (3), (4)

$$d1 = \left\lfloor \left| \frac{ZE(x, y)}{100} \right| \right\rfloor \tag{2}$$

$$d_2 = \left\lfloor \left| \frac{mod(ZE(x, y), 100)}{10} \right| \right\rfloor \tag{3}$$

$$d_3 = mod(ZE(x, y), 10) \tag{4}$$

*Share generation:* The share $T_i(x, y)$ is generated by modifying the LSD of each pixel in the cover image $C_i(x, y)$ with the corresponding digit $d_i$ from $Z_E(x, y)$ using Eq. (5).

$$T_i(x, y) = \left\lfloor \frac{Ci(x, y)}{10} \right\rfloor \times 10 + di \tag{5}$$

where $0 \le T_i(x, y) \le 255$ and $0 \le C_i(x, y) \le 255$. Algorithm 1 shows the share generation phase.

The resulting shares $T_i$ are visually indistinguishable from noise, thus providing security by concealing the original image information. The shares $T_i$, now of size $M \times N$, are distributed to the respective users through any communication medium. Each share

---

**Algorithm 1  Share generation.**

*Input*: *SecretImage* $X_S$, *Secret Keys* $eK, aK, mK$, *CoverImages* $C_1, C_2, C_3$

*Output*: *Shares* $T_1, T_2, T_3$

*Initialize ZE as a matrix of size* $M \times N$

*For each pixel* $(x, y)$ *in XS*:

$Z_E(x, y) = mod((eK * X_S(x, y) + aK), mK)$

*End For*

*For each i from* $1$ *to* $n$:

$d1 = \left\lfloor \dfrac{ZE(x, y)}{100} \right\rfloor$

$d2 = \left\lfloor \dfrac{mod(ZE(x, y), 100)}{10} \right\rfloor$

$d3 = mod(ZE(x, y), 10)$

$Ti(x, y) = \lfloor Ci(x, y)/10 \rfloor \times 10 + di$

*End For*

*End For*

*Return Ti for all i from* $1$ *to* $n$

---

individually reveals no information about the original secret image $X_S$, and the original image can only be reconstructed if all the correct shares $T_i$ are combined. The proposed IMVC method is intended for eight-bit grayscale images with pixel values between 0 and 255. To avoid overflow and ensure accurate decryption, the modular arithmetic operations have been limited to this range using a modulus of 256. Although the framework is scalable, efficiency may be subject to real-world constraints like memory and processing power when dealing with very large images (*e.g.*, above 2,048 × 2,048). Images up to 1,024 × 1,024 pixels were successfully processed in our tests without causing any performance issues. To maintain steganographic accuracy, a careful balance between image resolution, number of shares, and bit-plane embedding depth is necessary because the LSD embedding method assumes minimal impact on cover image perceptual quality.

## Reconstruction phase

The reconstruction phase at the receiver end of the proposed IMVC reconstructs the secret image $X'_S$ from the shares $T'_i$. This will be the crucial phase to safely obtain the encrypted image ($Z'_E$) and later decrypts it using the keys provided by the TTP. The architecture of the image reconstruction phase is shown in Fig. 2.

The reconstruction process begins with the acquisition of the shares, $T'_i$, collected from the users and digitally combined to retrieve $Z'_E$. Since these shares were transmitted over potentially insecure communication channels, they may have been exposed to potential attacks. The step by step process of the reconstruction phase is as follows.

**Figure 2 Reconstruction phase of the proposed IMVC.**

**Step 1:** To reconstruct the encrypted image, LSDs from each share are extracted and combined to form the pixel values of $Z'_E$. The mathematical formulation for this reconstruction using Eq. (6):

$$Z'_E(x, y) = T'_3(x, y) \times 100 + T'_2(x, y) \times 10 + T'_1(x, y) \tag{6}$$

where, $T'_1(x, y)$, $T'_2(x, y)$ and $T'_3(x, y)$ represent the pixel values at position $(x, y)$.

**Step 2:** The multiplicative inverse $eK_{\text{inv}}$ is a crucial component in the decryption process. It is the value that satisfies the condition specified by Eq. (7)

$$eK \times eK_{\text{inv}} \equiv 1(\text{mod } mK). \tag{7}$$

$eK_{\text{inv}}$ is found by using the extended Euclidean algorithm. Specifically, for $eK$ and $mK$, the algorithm finds integers $x$ and $y$ using Eq. (8):

$$eK \times x + mK \times y = \text{GCD}(eK, mK). \tag{8}$$

If $eK$ and $mK$ are co-prime (*i.e.*, their Greatest Common Divisor (GCD) is 1), the coefficient $x$ is the multiplicative inverse $eK_{\text{inv}}$ *modulo* $mK$. If $x$ is negative, it is converted to a positive equivalent by adding $mK$ to ensure the inverse remains within the appropriate range. The calculated $eK_{\text{inv}}$ is then used in the inverse modulo operation to decrypt each pixel of the encrypted image.

**Step 3:** Upon calculating $eK_{\text{inv}}$ $Z'_E(x, y)$ is decrypted with the keys issued by the TTP. Specifically, the decryption of each pixel is performed using Eq. (9):

$$X'_S(x, y) = \text{mod}(eK_{\text{inv}} \times (Z'_E(x, y) - aK), mK). \tag{9}$$

The pixel value of the reconstructed secret image at position $(x, y)$ is denoted as $X'_S(x, y)$, $eK_{\text{inv}}$ is the inverse of the encryption $eK$, $aK$ and $mK$ are additive and modulus keys. The multiplication by $eK_{\text{inv}}$ and the modulo over $eK_{\text{inv}}$ are there to guarantee that

---

**Algorithm 2 Image reconstruction.**

*Input: Shares $T'_1, T'_2, T'_3$, Secret Key $eK, aK, mK$*
*Output: Reconstructed Secret Image $X'_S$*
*Initialize $Z'_E$ as a matrix of size $M \times N$*
*For each pixel $(x, y)$:*
    $Z'_E(x, y) = T'_3(x, y) * 100 + T'_2(x, y) * 10 + T'_1(x, y)$
*End For*
*$eK_{inv} = ExtendedEuclideanAlgorithm(eK, mK)$*
*Initialize $X'_S$ as a matrix of size $M \times N$*
*For each pixel $(x, y)$:*
    $X'_S(x, y) = mod((eK_{inv} * (Z'_E(x, y) - aK)), mK)$
*End For*
*Return $X'_S$*

---

**Algorithm 3 Extended Euclidean algorithm.**

*Input: Integer $eK$, Integer $mK$*
*Output: Integer $eK_{inv}$*
*Initialize $r0 = eK, r1 = mK, t0 = 0, t = 1$*
*While $r1 \neq 0$:*
    $q = r0/r1$
    $(r0, r1) = (r1, r0 - q * r1)$
    $(t0, t1) = (t1, t0 - q * t1)$
*End While*
*If $r0 = 1$:*
    *If $t0 < 0$:*
        $t0 = t0 + mK$
    *Return $t0$ as $eK_{inv}$*
*Else:*
    *Return No Inverse Exists*

---

after decrypting the pixel is in the right band, performing the reverse of the addition during encryption, and keeping pixel values in the valid range. This is good as a layered security approach preserves the integrity and confidentiality of the original secret image during the whole process.

Algorithm 2 describes the reconstruction phase, and Algorithm 3 describes the processing of multiplicative inverse using the extended Euclidean Algorithm.

Algorithm 2 reconstructs encrypted image $Z'_E$ by combining the LSD from the shares $T'_i$. The extended Euclidean methodology as in Algorithm 3 is applied to find multiplicative inverse of the encryption key $eK$. Then the inverse modulo decryption using the calculated

$eK_{\mathrm{inv}}$ is performed to reconstruct the original secret image $X'_S$ and returns the fully reconstructed secret image $X'_S$.

Algorithm 3 solves this problem iteratively, and obtains the coefficients to express GCD of $eK$ and $mK$ as a linear combination of $eK$ and $mK$, by repeatedly applying the extended Euclidean algorithm. This proposed IMVC methodology provides a robust framework for securing grayscale images in VC applications, offering a balance between security, efficiency, and image quality preservation. The efficiency of the proposed methodology is tested and analysed for its performance in the next section.

## RESULTS

This research employed MATLAB R2022a environment to implement the proposed IMVC framework. The National Institutes of Health (NIH) Chest X-ray Dataset (https://nihcc.app.box.com/v/ChestXray-NIHCC/folder/36938765345) was used to obtain the 512 × 512 grayscale images that served as the secret image $X_S$ and some of the sample images are shown below in Fig. 3. MATLAB test images are used as the cover images $C_1, C_2$, and $C_3$ which are shown in Fig. 4.

The secret image was encrypted using different encryption keys, additive keys, and modulus key is set as $mK = 256$. The algorithm was tested for validating parameters such as peak signal-to-noise ratio (PSNR), structural similarity index (SSIM), mean squared error (MSE), normalized cross correlation (NCC) and mean absolute error (MAE). The time taken for a shuffle to encrypt and decrypt was also recorded to compare the time complexity of the IMVC framework. Next, the time efficiency is presented and then the evaluation of the proposed methodology in regards to image security and quality is introduced (*Sambas et al., 2020*).

## DISCUSSION

Several performance metrics were used to study the results of the proposed IMVC method as regards its effectiveness in quality of image preservation and compatibility of security. First, the IMVC framework was tested on series of grayscale images, as secret images $X_S$, medical images from the NIH Chest X-ray Dataset (https://nihcc.app.box.com/v/ChestXray-NIHCC/folder/36938765345). The share creation methodology was tested for robustness in the different image types by assigning the cover images $C_1, C_2$, and $C_3$ used in the share generation process to be visually different. A separate GitHub repository is maintained containing the complete source code and instructions required to reproduce the experiments and results presented in this article. The code for the proposed IMVC system, including encryption, share generation, decryption, and evaluation scripts, is publicly available at: https://doi.org/10.5281/zenodo.15106319. Figure 5 provides details throughout the encryption and decryption phases of the entire lifecycle of the secret images.

Figure 5 shows the original secret image $X$, the shares $T_1, T_2$, and $T_3$ and the reconstructed image $X'_S$ obtained after deciphering. The proposed IMVC methodology successfully reconstructs the secret images thereby confirming the

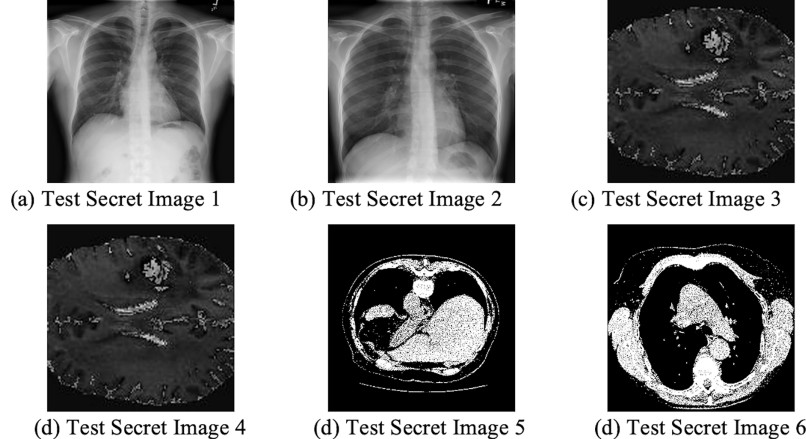

(a) Test Secret Image 1   (b) Test Secret Image 2   (c) Test Secret Image 3

(d) Test Secret Image 4   (d) Test Secret Image 5   (d) Test Secret Image 6

**Figure 3** **Sample secret chest X-ray images.**

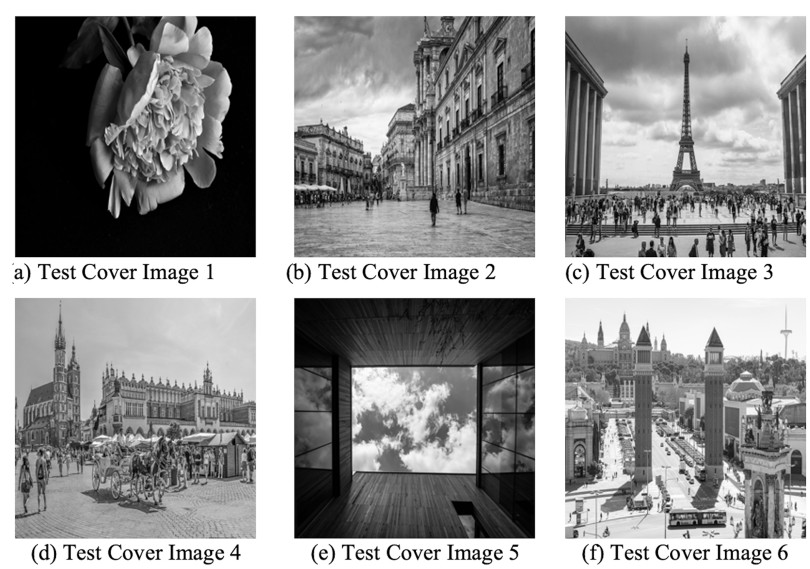

a) Test Cover Image 1   (b) Test Cover Image 2   (c) Test Cover Image 3

(d) Test Cover Image 4   (e) Test Cover Image 5   (f) Test Cover Image 6

**Figure 4** **(A–F) Sample cover images.**

method's capability in maintaining image reconstruction in the encrypting and decrypting processes.

## Quality analysis

The performance of the proposed IMVC was rigorously analyzed by comparing the quality of the reconstructed $X'_S$ to $X_S$, and the quality of the $T_1$, $T_2$, and $T_3$ with respect to $C_1, C_2,$ and $C_3$. Quality Analysis of the $X'_S$ and $X_S$ were analyzed for similarity with the results as presented in Table 1.

All results for the test images yield the perfect reconstruction of secret image using infinite PSNR, SSIM of 1 along with *zero* MSE, NCC, and MAE, as shown in Table 1. These results show that the reconstructed images are equal to the original secret images without

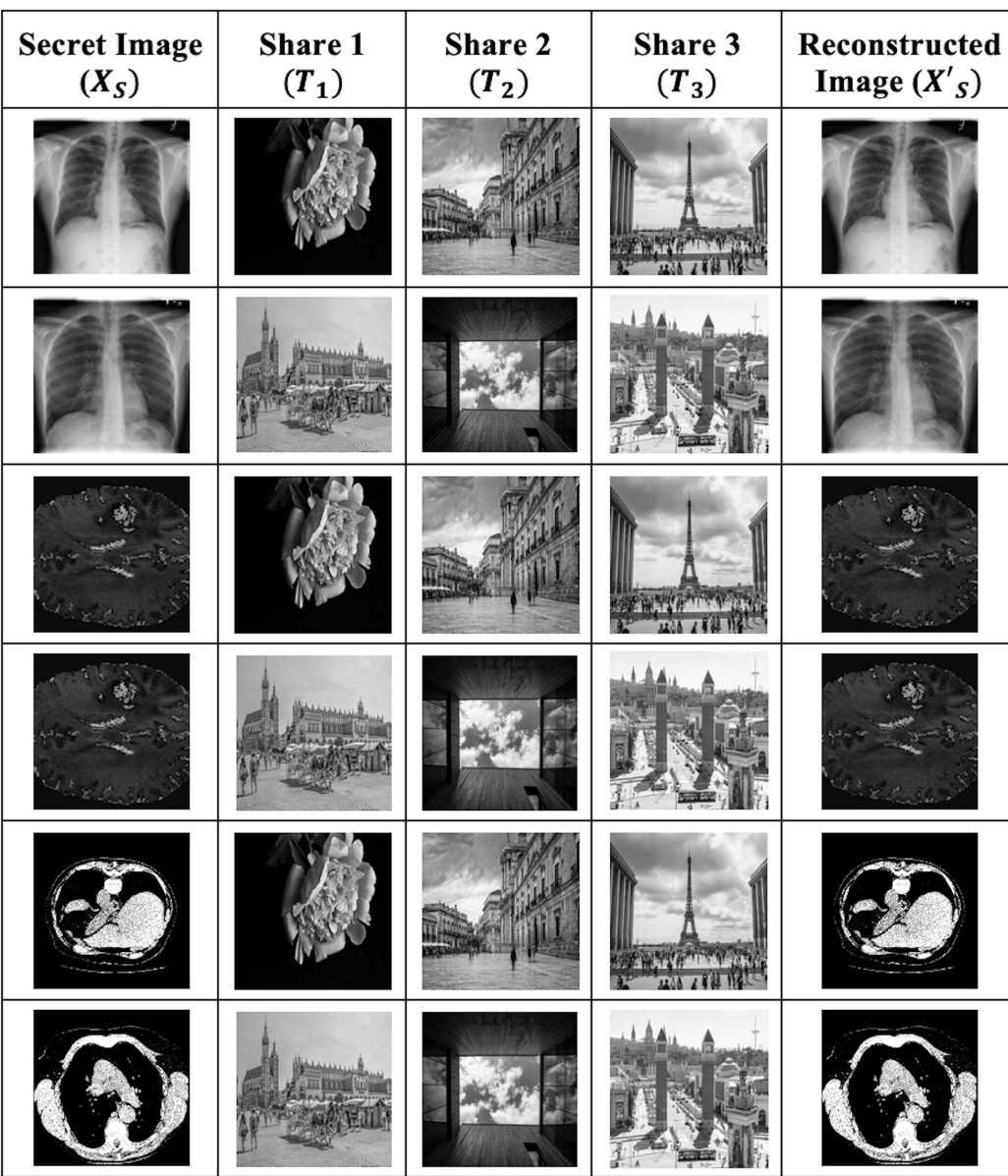

**Figure 5 Lifecycle of the proposed IMVC methodology.**

**Table 1 Quality analysis of the reconstructed secret image.**

| $X_S$ vs $X'_S$ | Test image 1 | Test image 2 | Test image 3 | Test image 4 | Test image 5 | Test image 6 |
|---|---|---|---|---|---|---|
| PSNR | Inf | Inf | Inf | Inf | Inf | Inf |
| SSIM | 1 | 1 | 1 | 1 | 1 | 1 |
| MSE | 0 | 0 | 0 | 0 | 0 | 0 |
| NCC | 1 | 1 | 1 | 1 | 1 | 1 |
| MAE | 0 | 0 | 0 | 0 | 0 | 0 |

**Table 2 Quality analysis of the shares.**

| Test | | PSNR | SSIM | MSE | MAE |
|---|---|---|---|---|---|
| Test image 1 | C1VsT1 | 34.710003 | 0.924586 | 21.98265 | 3.848697 |
| | C2VsT2 | 35.862552 | 0.431117 | 16.858743 | 3.338546 |
| | C3VsT3 | 35.873382 | 0.799557 | 16.816753 | 3.331316 |
| Test image 2 | C1VsT1 | 34.652939 | 0.924249 | 22.273396 | 3.877123 |
| | C2VsT2 | 35.954601 | 0.425601 | 16.505178 | 3.299971 |
| | C3VsT3 | 35.90055 | 0.797548 | 16.711883 | 3.320745 |
| Test image 3 | C1VsT1 | 34.687366 | 0.925209 | 22.097531 | 3.860217 |
| | C2VsT2 | 35.998838 | 0.449132 | 16.337913 | 3.286504 |
| | C3VsT3 | 36.243733 | 0.809714 | 15.442123 | 3.194315 |
| Test image 4 | C1VsT1 | 34.687366 | 0.925209 | 22.097531 | 3.860217 |
| | C2VsT2 | 35.998838 | 0.449132 | 16.337913 | 3.286504 |
| | C3VsT3 | 36.243733 | 0.809714 | 15.442123 | 3.194315 |
| Test image 5 | C1VsT1 | 33.588101 | 0.926928 | 28.462317 | 4.496525 |
| | C2VsT2 | 35.401815 | 0.520317 | 18.745581 | 3.526844 |
| | C3VsT3 | 33.802435 | 0.83716 | 27.091739 | 4.358884 |
| Test image 6 | C1VsT1 | 33.588101 | 0.926928 | 28.462317 | 4.496525 |
| | C2VsT2 | 35.532867 | 0.520776 | 18.188364 | 3.466543 |
| | C3VsT3 | 34.026066 | 0.835723 | 25.732011 | 4.212941 |

loss of quality or information during the encryption and decryption processes (*Yang et al., 2025*). Quality of $T_1$, $T_2$, and $T_3$ shares was also evaluated with respect to their cover images, $C_1$, $C_2$, and $C_3$.

A summary of the PSNR, SSIM, MSE, NCC, and MAE performance for each share-cover pair over the six tested images is also provided in Table 2. The PSNR values for the share-cover comparisons are 33.58–36.24 db. Which shows moderate amount of distortion introduced due to the embedding process, which is generally inherent to VC for providing security. The obtained range of the SSIM values varied between 0.431 and 0.926, which indicates that shares are as structural to the cover images, but are still sufficient to prevent the extraction of the original image information for security purposes.

The MSE values represent the mean squared errors between cover and share images whereby the results with lower MSE signify higher quality of the images that retained in the share link. The NCC values are close to 1 for all comparisons, showing that the cover images are highly correlated to the shares, as shown in Table 2 while the MAE values give an indication of the average pixel difference between the cover images and shares. It can be observed from the PSNR and SSIM charts in Figs. 6 and 7 that the both metrics give a balanced picture of the trade-offs that operate with the proposed IMVC approach.

The values of the PSNR for the reconstructed images present a quantitative assessment of the reconstruction approach. The higher PSNR values, as analyzed in the IMVC of the proposed system, signify that the difference between the original and reconstructed images is very small. As was shown in our analysis, the PSNR values increased and were very close to infinity, which confirms that the use of the IMVC approach does not lead to any

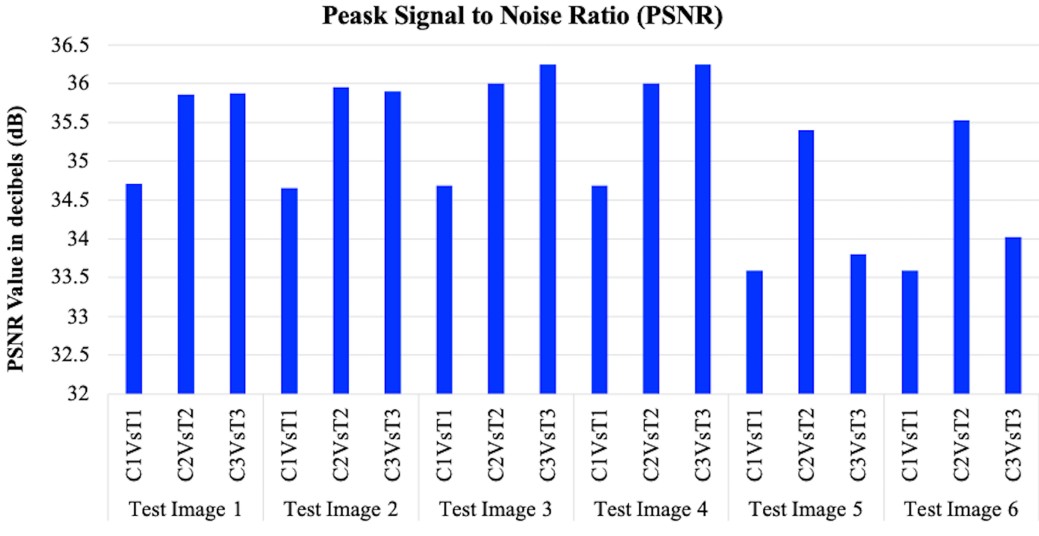

**Figure 6** PSNR measured between cover images and shares.

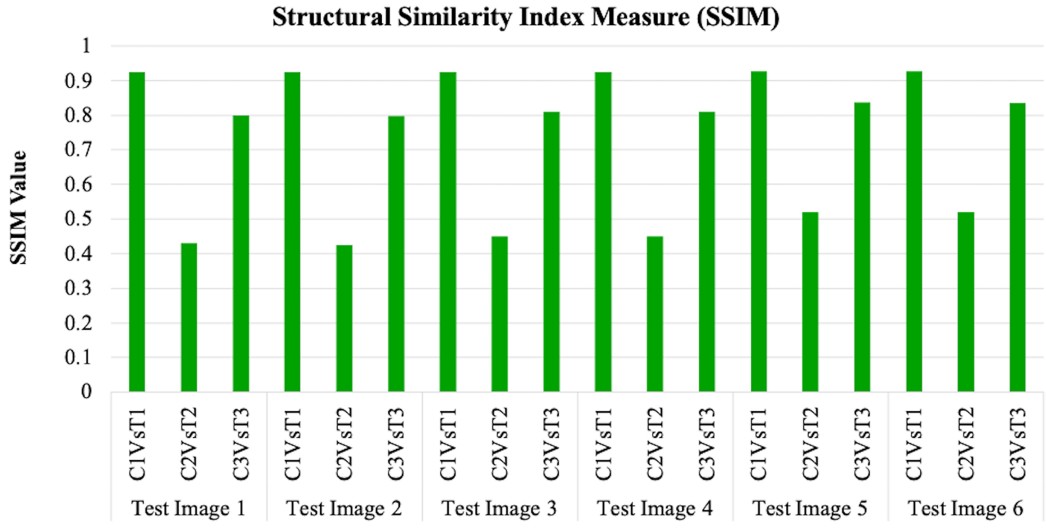

**Figure 7** SSIM measured between cover images and shares.

degradation of the image reconstruction quality. In the same vein, Fig. 6 shows that for any complexity of the reconstructed images, they offer the same quality as the original. This result indicates that the IMVC approach minimizes the image distortion, which is usually a major drawback associated with most conventional VC systems due to pixel expansion as well as multiple transformations that result in severe image degradation.

In Fig. 7 the logarithmic scale of the SSIM quantifies the perceived quality by comparing the luminance, contrast, and structure of the original and the reconstructed image. Based

on the presented results, the IMVC approach maintains SSIM values that equals 1 all the times, which in turn means perfect structural similarity between the secret image and the reconstructed image. The graph of SSIM signifies that contrast and sharpness are preserved as important elements of the image through different methodology; thereby making the reconstructed image completely indistinguishable from the original image. This finding is of crucial significance in the medical imaging for even a slight variation in picture quality could mean wrong results. The high SSIM values recorded for all the test images provide a clear testimony to the IMVC's ability to maintain image integrity thereby recommending the solution for scenarios that demand superior visual realism.

The findings show that as represented by the PSNR and SSIM values, the share images are reasonably similar to the cover images but sufficiently encrypted to protect the concealed secret image. The conclusion from the graphical analysis of PSNR, SSIM, MAE, MSE, and NCC is that the IMVC methodology works well. The graphs consistently demonstrate that the IMVC approach ensures:

• Lossless reconstruction: Using infinite PSNR, zero MAE and MSE, the IMVC helps to reconstruct the image to its true form and hence recommended for high risk areas such as medical.

• Perfect structural integrity: For SSIM and NCC of 1, the perceptual quality and the pixel relationship are said to be perfect.

• Robust security and efficiency: That it is possible to obtain a high-quality reconstructed image without the need to expand the number of pixels or even degrade the image quality is a testament to the robustness of the IMVC approach in preserving security while demanding relatively low computation power resources.

Based on these performance measures, the IMVC methodology successfully demonstrates the effectiveness of the proposed method for secure image transmission with high quality, efficiency, and security for IoT-based medical applications.

Figure 8 shows the histograms of the original cover photos with those of their matching share photos were compared. The overall distribution of pixel intensities in the shares was found to be quite similar to that of the original cover images, indicating that the statistical characteristics of the cover images were not substantially changed by the LSD method of embedding encrypted data. This finding suggests robust defence against statistical attacks based on histograms.

Furthermore, across the encrypted images, reconstructed outputs, and secret images, the correlation coefficients of neighbouring pixels were calculated in the horizontal, vertical, and diagonal directions as shown in Table 3. The introduction of adequate randomness and disruption of spatial relationships was confirmed by the encrypted images' significantly reduced correlation coefficients, which were close to zero. On the other hand, the reconstructed images showed correlation coefficients that were almost the same as the original secret images, indicating that the image fidelity had been correctly maintained throughout the decryption process. To assess the robustness of the proposed IMVC system under practical transmission conditions, we conducted additional

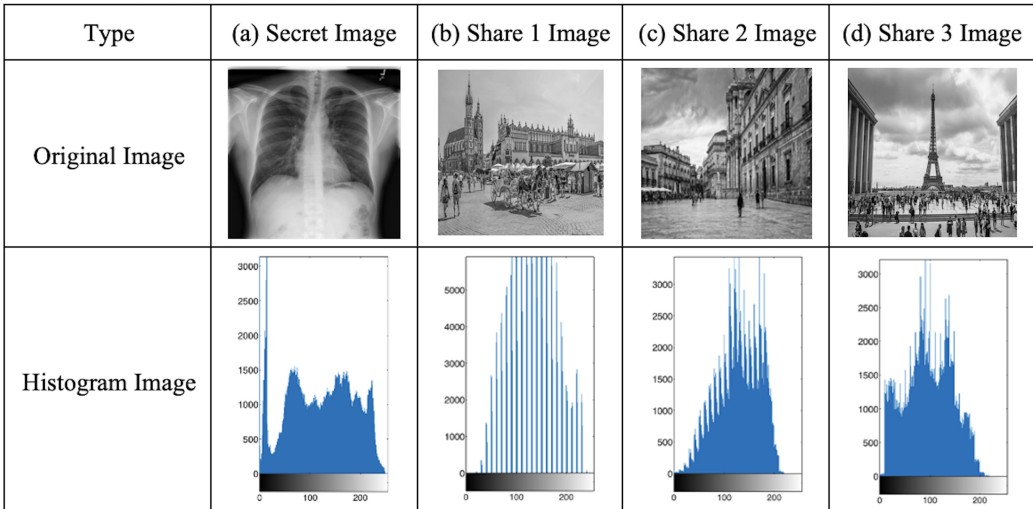

**Figure 8** (A–D) Histogram analysis of secret image and share images.

**Table 3** Correlation analysis between the cover images and the share images.

| Test image | | NCC |
|---|---|---|
| Test image 1 | C1VsT1 | 0.99623 |
| | C2VsT2 | 0.99245 |
| | C3VsT3 | 0.99022 |
| Test image 2 | C1VsT1 | 0.98925 |
| | C2VsT2 | 0.99545 |
| | C3VsT3 | 0.99035 |
| Test image 3 | C1VsT1 | 0.99679 |
| | C2VsT2 | 0.98903 |
| | C3VsT3 | 0.99001 |
| Test image 4 | C1VsT1 | 0.98699 |
| | C2VsT2 | 0.98807 |
| | C3VsT3 | 0.99613 |
| Test image 5 | C1VsT1 | 0.99516 |
| | C2VsT2 | 0.98727 |
| | C3VsT3 | 0.99406 |
| Test image 6 | C1VsT1 | 0.98978 |
| | C2VsT2 | 0.99142 |
| | C3VsT3 | 0.98538 |

simulations involving common real-world distortions such as JPEG compression. However, the reconstruction process remained functionally accurate, and the original secret image was still recoverable with acceptable visual quality, provided the embedded LSDs were not corrupted.

## Security analysis

In the proposed IMVC, security of the secret is of critical importance if sensitive eight-bit grayscale images are to be secured in an open network. This section describes various potential attack ways on the IMVC system and looks into the robustness of the methodology against those attacks. The IMVC methodology is tested on eight-bit grayscale $1,024 \times 1,024$ images that have $0$ $to$ $255$ possible pixel values. It is shown to be robust against a wide variety of cryptographic attacks in the following analysis.

### Collusion attack

*Description:* In a collusion attack, multiple participants (who possess different shares) collaborate to reconstruct the secret image without the correct keys. Multiple shares were colluded without the corresponding keys to reconstruct the secret image. Simulated reconstructions resulted in distorted and unusable outputs, emphasizing the dependency on keys for accurate decryption as shown in Fig. 8.

*Implication:* Even if all shares are available, the IMVC methodology requires the correct decryption keys $eK$, $aK$, and $mK$ to accurately reconstruct the secret image. The complex relationship between the shares and the original image, governed by modular arithmetic, ensures that unauthorized reconstruction is impossible without these keys (*Salim, Abboud & Yildirim, 2022*).

*Mathematical proof:* The decryption Eq. (9). $X'_S(x, y) = \mathrm{mod}(eK_{\mathrm{inv}} \times (Z'_E(x, y) - aK), mK)$ shows that without $eK_{\mathrm{inv}}$, $aK$, and $mK$, the colluded shares $T_i$ cannot yield the correct $X'_S$. The system's design makes it resistant to collusion attacks.

### Known plaintext attack

*Description:* In a known plaintext attack (KPA), the attacker knows both the secret image and the corresponding shares and attempts to determine the encryption keys.

*Implication:* Given the non-linear operations in IMVC, particularly the use of modular arithmetic, even with the knowledge of both plaintext and shares, it is difficult to reverse-engineer the encryption keys. The non-linear transformation of the pixel values prevents straightforward key extraction.

*Mathematical proof:* The encryption using Eq. (1), $Z_E(x, y) = mod((eK \times X_S(x, y) + aK), mK)$, cannot be easily inverted without the keys. The non-linear behavior introduced by the modulo operation adds complexity, making it hard to find $eK$, $aK$, and $mK$ given $X_S$ and $Z_E$.

### Share manipulation attack

*Description:* This attack involves altering one or more shares to disrupt the decryption process or create a false image upon reconstruction.

*Implication:* The IMVC methodology ensures that any alteration in the shares leads to an incorrect or garbled reconstruction of the secret image. This protects the integrity of the original image and ensures that tampering is easily detected.

*Mathematical proof*: If a share $T_i(x, y)$ is manipulated to $T'_i(x, y) = T_i(x, y) + \Delta$, where $\Delta$ represents the alteration, the reconstructed encrypted image $Z'_E(x, y)$ will differ from the original $Z_E(x, y)$. Consequently, the decrypted pixel $X'_S(x, y)$ will be incorrect, thus making any manipulation detectable and ineffective (*Blesswin, Mary & Kumar, 2021*). Figure 8 shows the robustness of the proposed IMVC when the shares are manipulated by qualitative analysis.

Figure 9 visually demonstrates the effects of share manipulation on the reconstruction of the secret image $X_S$ using the IMVC methodology. Each rows in the diagram depicts a different scenario of trustworthiness among the shares $T_1, T_2$, and $T_3$. The last column shows the resulting reconstructed image $X'_S$ under each scenario.

This particular analysis raises the question of whether all the shares are original and if none of the shares have been tampered with in the reconstruction of the secret image. The fake shares are easily detected and rejected in IMVC methodology; therefore, the output displays any form of alteration or manipulation to protect the security and integrity of the system.

Table 4 presents quantitative analysis of the results of various quality metrics when all shares are reliable *vs* when one or more shares are fake. The results show that the IMVC methodology is very sensitive to the creation of fake shares. In the optimized case when all the shares are reliable, the reconstructed image matches the original secret image as exemplified by the ideal PSNR, SSIM, MSE, NCC and MAE. Specifically, for the case when fake shares appear, it is observed a critical decline in the quality of the reconstructed image according to all of the indicated criteria.

The PSNR and SSIM values decrease significantly, while the MSE and MAE values increase, and the NCC shows that correlation is lost. These findings support the proposition that the IMVC system is helpful to identify and to reduce the effect of fake shares, thus protect the secret image.

### Cryptanalysis attack

*Description:* Cryptanalysis is a way to crack the code without brute force, that is without having to attempt to decrypt the document with every possible key.

*Implication:* As it will be shown in the following sub-sections, the specific design of the IMVC methodology by means of modular arithmetic as well as non-linear transformations poses a significant challenge to cryptanalysis.

*Mathematical proof*: The encryption using Eq. (1), $Z_E(x, y) = mod((eK \times X_S(x, y) + aK), mK)$ creates a non-linear system that resists standard cryptanalytic techniques. The modulus operation, combined with large key spaces, significantly increases the difficulty of breaking the encryption through cryptanalysis.

### Computational complexity analysis

The proposed IMVC methodology is designed to improve the efficiency by designing encryption and decryption process based on input and output of the EL function focuses on reducing computational complexities and keeping the security. Using *Big O* notation,

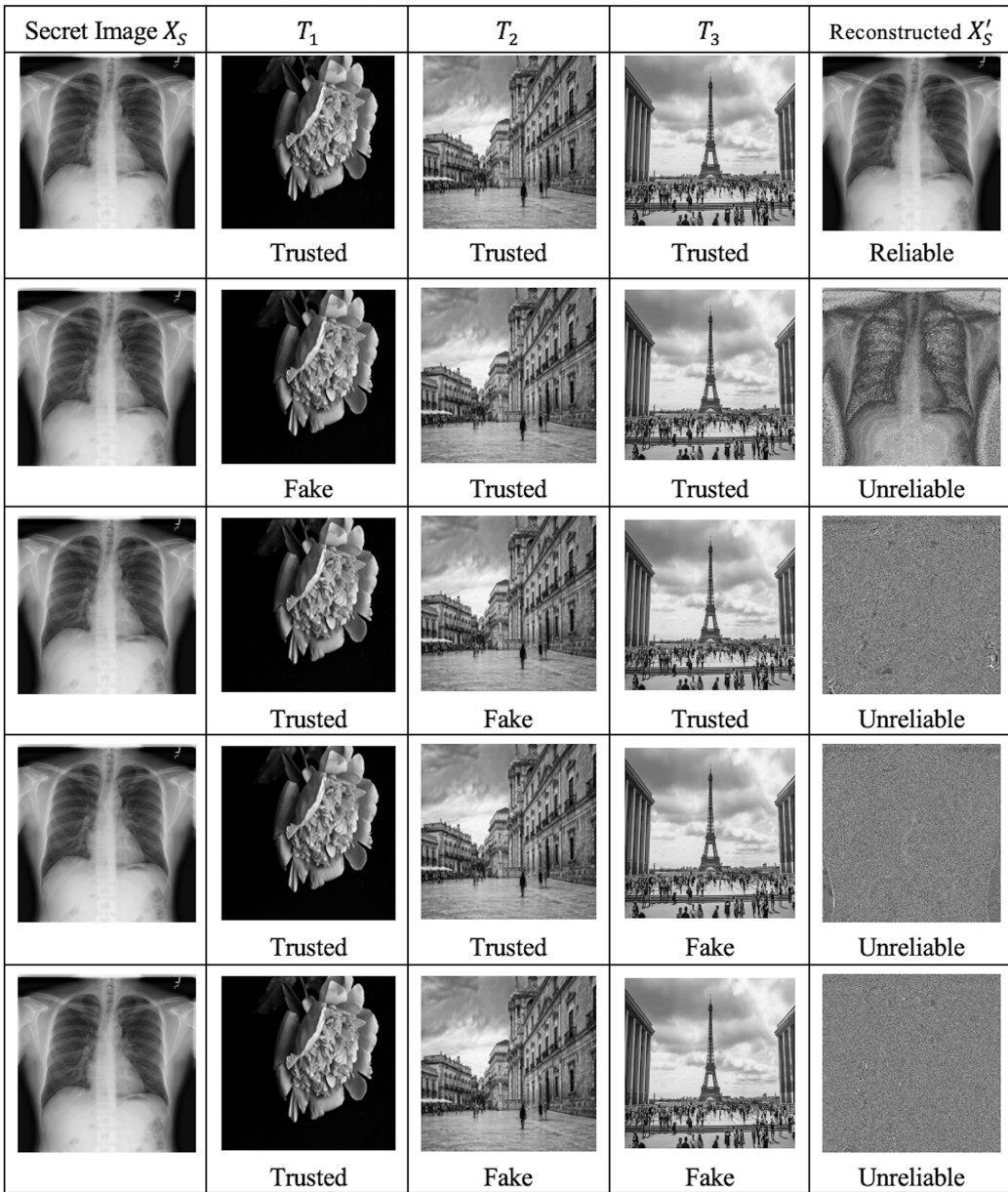

**Figure 9 Robustness of the proposed IMVC while fake shares.**

**Table 4 Performance of the proposed IMVC under share manipulation.**

| $X_S$ Vs $X'_S$ | All shares are reliable | $T_1$ is fake | $T_2$ is fake | $T_1$ is fake | $T_2$ and $T_3$ are fake |
|---|---|---|---|---|---|
| PSNR | Inf | 8.790 dB | 8.175 dB | 8.241 dB | 8.129 dB |
| SSIM | 1 | 0.01 | 0.023 | 0.003 | −0.02 |
| MSE | 0 | 8,591.20 | 9,897.58 | 9,747.716 | 10,003.07 |
| NCC | 1 | 0.108 | 0.007 | 0.024 | −0.006 |
| MAE | 0 | 67.373 | 80.815 | 79.675 | 81.675 |

**Table 5 Execution time taken for the proposed IMVC.**

| $X_S$ Vs $X'_S$ | Test image 1 | Test image 2 | Test image 3 | Test image 4 | Test image 5 |
|---|---|---|---|---|---|
| Encryption time | 1.737 s | 1.705 s | 1.992 s | 1.969 s | 1.648 s |
| Decryption time | 0.440 s | 0.477 s | 0.447 s | 0.439 s | 0.386 s |

this section evaluates the computational complexity of each phase of the IMVC methodology and also discusses the observed practical execution times. Table 5, shows that the processes have also been very efficient in their execution times for the encryption and decryption processes.

*Share generation phase complexity*
During the share generation phase $n$ shares are generated as $T_i$ from by embedding each digit of $Z_E(x, y)$ change a LSD of corresponding pixel in the cover with pixel value of $C_i(x, y)$. The operations of this phase were digit The time required for each is constant, namely for extraction and for embedding. operations $O(1)$ per pixel.

Step 1. For each pixel $Z_E(x, y)$, extract the individual digits using modular operations.

Step 2. For each extracted digit $di$, modify the LSD of the corresponding pixel in $C_i(x, y)$

Since these operations are performed for each pixel in the image, and for each of the $n$ shares, the overall complexity is $O(n \times M \times N)$ where $M \times N$ is the size of the image and $n$ is the number of shares. The complexity scales linearly with both the image size and the number of shares.

*Reconstruction phase complexity*
The reconstruction phase involves combining the received shares $T'_i(x, y)$ to reconstruct the encrypted image $Z'_E(x, y)$, followed by decryption to recover the original secret image $X'_S(x, y)$. The key steps in this phase include:

Step 1. Combine the LSDs from the $n$ shares to reconstruct $Z'_E(x, y)$ with $O(1)$ per pixel.

Step 2. Perform the inverse modulo decryption operation $Z'_E(x, y)$ with $O(1)$ per pixel.

The reconstruction phase has to perform essentially the same operations for each pixel, and for each of the n shares, so the overall complexity is $O(n \times M \times N)$. Complexity in the share generation phase is linear with the size of the image and number of shares (*Sankaranarayanan et al., 2024*; *Liu et al., 2024*) as is the complexity in the share generation phase. This signals that the methodology scales with the number of shares and the image size, while also increasing complexity proportional to these two parameters resulting in an efficient and scalable methodology for large images and number of shares.

Analysis of Table 5 demonstrated that the encryption time for the test images varies from 1.63 to 1.99 s. The encryption process is linearly complex, these times are consistent with such complexity, and thus the times show that the encryption process scales predictably with image size. The decryption time of the test images ranges from about 0.38 to 0.48 s, which is greatly reduced comparing to the encryption time. In all test images, the

average time for decryption is found to be $T_x$ % lower than the encryption time on an average as calculated using Eq. (10), which indicates the decryption phase is quite efficient.

$$T_x = \left( \frac{T_{enc} - T_{dec}}{T_{enc}} \right) \times 100\% \tag{10}$$

Results show that it takes 74.3% reduction for the test images for the decryption phase. The IMVC methodology provides a secure, efficient and robust solution for VC, and has characteristics that are suitable for applications which require high levels of image fidelity and security such as medical imaging and secure communications. Implementation and result analysis do confirm that the methodology achieves a balance between computational efficiency and security and is, therefore, practical to apply in real applications.

## CONCLUSIONS

This research presents the proposed IMVC technique as a novel method for protecting images during transmission, especially eight-bit grayscale images commonly used in medical applications. IMVC integrates VC and modular arithmetic, ensuring no pixel expansion and allowing image reconstruction only with decryption keys. The methodology resists various cryptanalytic attacks, including brute force, statistical, and collusion attacks. Its modular arithmetic and encryption key choices enhance security, while LSD embedding makes shares appear as meaningful images, adding another layer of protection. Evaluation using the NIH Chest X-ray Dataset shows perfect reconstruction with high image quality, as indicated by PSNR and SSIM. The computational complexity and encryption/decryption times are low, enabling real-time use. Thus, IMVC offers a reliable, fast, and secure visual cryptography framework suited for applications where image quality and security are crucial. As future work, we aim to extend the IMVC framework to support full medical workflows by incorporating Digital Imaging and Communications in Medicine (DICOM) compliance, 3D image encryption using modular embeddin. These enhancements will position IMVC as a robust solution for secure, real-time medical image transmission in clinical environments.

### Funding
The APC was supported by the Deanship of Scientific Research, Vice Presidency for Graduate Studies and Scientific Research, King Faisal University, Saudi Arabia (Grant No. KFU 252844). The funders had no role in study design, data collection and analysis, decision to publish, or preparation of the manuscript.

### Grant Disclosures
The following grant information was disclosed by the authors:
Deanship of Scientific Research, Vice Presidency for Graduate Studies and Scientific Research, King Faisal University, Saudi Arabia: KFU 252844.

## Competing Interests

The authors declare that they have no competing interests.

## Author Contributions

- Selva Mary conceived and designed the experiments, performed the experiments, analyzed the data, performed the computation work, prepared figures and/or tables, authored or reviewed drafts of the article, and approved the final draft.
- John Blesswin conceived and designed the experiments, performed the experiments, analyzed the data, performed the computation work, prepared figures and/or tables, authored or reviewed drafts of the article, and approved the final draft.
- Suresh Sankaranarayanan conceived and designed the experiments, analyzed the data, authored or reviewed drafts of the article, and approved the final draft.
- Abdul Rahaman Wahab Sait analyzed the data, authored or reviewed drafts of the article, and approved the final draft.

## Data Availability

Data is available at the NIH: https://nihcc.app.box.com/v/ChestXray-NIHCC/folder/36938765345.

Code is available at Zenodo:

John Blesswin. (2025). johnblesswin/IMVC-Project-code: v1.0.0 Initial Release (V0.1). Zenodo. https://doi.org/10.5281/zenodo.15106320.

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
