# Peer review of "Modular inverse visual cryptography for balancing security, quality, and efficiency in image transmission"

_PeerJ Computer Science, doi:10.7717/peerj-cs.3140_

## Round 0.1 · original submission · Major Revisions

Please take precautions to go through all criticisms.

Reviewer 1 ·

Basic reporting

The authors propose a new reversible visual cryptosystem for secure and efficient transmission of medical images in the Internet of Things. It distinguishes itself from traditional image encryption algorithms by using secret image sharing techniques and information hiding to generate visually secure cipher images, which brings good security. However, there are still some problems to be solved.

(1) Many of the references in the manuscript are too old, and it is recommended that they be replaced with the latest references, e.g., 2024, 2025. In addition, check for missing or inconsistent formatting of reference information.

(2) Is there a limit to the use of this method during the shared generation phase? For example, image size, range of pixel values, etc.

(3) Is the anomaly symbol in the paragraph following Table 1 a writing error?

Experimental design

-

Validity of the findings

(4) Add histogram analysis and correlation analysis.

(5) You may add comparative analysis with similar algorithms. For example,
a. doi.org/10.1016/j.chaos.2023.114111
b. doi.org/10.1007/s00530-025-01765-x
c. doi.org/10.1016/j.chaos.2024.115632

Reviewer 2 ·

Basic reporting

The manuscript is well-written in clear, professional English, with only minor grammatical and formatting issues that do not hinder comprehension. The introduction effectively contextualizes the need for secure image transmission in IoT-based healthcare, and the motivation for the proposed Inverse Modular Visual Cryptography (IMVC) system is well articulated. Relevant prior work is cited appropriately, and the literature review provides a solid theoretical foundation. Figures and tables are of high quality, clearly labeled, and support the narrative effectively, while the inclusion of publicly available implementation code and datasets ensures transparency and reproducibility. The manuscript follows PeerJ’s structural guidelines, with a logical flow from introduction to conclusion. Minor improvements in formatting, figure-text alignment, and proofreading would further enhance clarity and presentation.

Experimental design

This is my comment
1. Could the authors elaborate on how key selection (e.g., modulus size) impacts security
and decryption fidelity in resource-constrained devices?

2. Could real-world noise (e.g., transmission distortion or compression artifacts) affect
these perfect scores? If so, have simulations been tested?

3. Did the authors explore the visual detectability or steganalysis resilience of these
shares? Could such attacks distinguish between cover from share images?

Validity of the findings

This is my comment
1. Could partial manipulation of a share (not a full fake) still degrade output quality? How
tolerant is the reconstruction process to minimal corruption?

2. It would be useful to highlight whether the GitHub code includes visualization outputs
like PSNR/SSIM plots and fake share detection.

3. For future work, consider applying this scheme to full medical workflows (e.g.,
DICOM files, PACS systems) and extending to 3D imaging datasets.

---

## Round 0.2 · accepted · Accept

The authors properly addressed the requests and suggestions of the reviewers.

Reviewer 2 ·

Basic reporting

in the new version, article was very improved

Experimental design

in the new version, article was very improved

Validity of the findings

in the new version, article was very improved